# Quantitative Monitoring of Cyclic Glycine–Proline in Marine Mangrove-Derived Fungal Metabolites

**DOI:** 10.3390/jof10110779

**Published:** 2024-11-10

**Authors:** Jing Lin, Fei Qin, Zeye Lin, Weijian Lin, Minxin You, Li Xu, Lei Hu, Yung-Husan Chen

**Affiliations:** 1Xiamen Key Laboratory of Marine Medicinal Natural Product Resources, Xiamen Medical College, Xiamen 361023, China; 12462110034@fafu.edu.cn (J.L.); 201500010325@xmmc.edu.cn (F.Q.); linzeye00@outlook.com (Z.L.); linweijian@xmmc.edu.cn (W.L.); 202205480025@xmmc.edu.cn (M.Y.); xli@xmmc.edu.cn (L.X.); 2Haixia Institute of Science and Technology, Fujian Agriculture and Forestry University, Fuzhou 350002, China; 3Engineering Research Centre of Marine Biopharmaceutical Resource, Fujian Province University, Xiamen Medical College, Xiamen 361023, China

**Keywords:** UPLC-MS/MS, cyclic glycine–proline, *Penicillium pedernalense*, quantitative approach

## Abstract

This study developed and validated a robust UPLC-MS/MS method for quantifying cyclic glycine–proline (cGP) in mangrove-derived *Penicillium* and *Aspergillus* strains. The method demonstrated excellent linearity, precision, and recovery, with detection limits as low as 4.8 ng/mL. *Penicillium pedernalense* extract achieved a cGP content of 67.45 ± 1.11 ng/mL, with a corresponding fermentation yield of 29.31 ± 0.61 mg/L. This surpassed *Penicillium steckii*, which reached a content of 31.71 ± 0.31 ng/mL, with a yield of 8.51 ± 0.15 mg/L. This quantitative approach for metabolite analysis provides a viable method for screening these fungal strains, highlighting their potential for sustainable production of cyclic glycine–proline (cGP).

## 1. Introduction

Cyclic glycine–proline (cGP, 1, Figure 1) is a widely studied marine-derived cyclic peptide characterized by a conformationally constrained diketopiperazine (DKP) core [1,2]. Known for its neuroprotective properties, cGP has shown great potential for therapeutic applications, particularly in treating neurological disorders by modulating insulin-like growth factor 1 (IGF-1) homeostasis [3]. The pharmaceutical promise of cGP is further exemplified by its inclusion in drug candidates like NNZ-2591 (2, Figure 1), which is currently in Phase II clinical trials for treating Prader–Willi syndrome and Angelman syndrome, underscoring its importance in modern drug development [4]. In addition, cGP is utilized in nutraceutical products such as cGPMAX™ for cognitive and cardiovascular health in older adults [5]. Beyond its role in IGF-1 regulation, cGP is a valuable building block for preclinical drug candidates (Compounds 3–5, Figure 1) with a wide range of biological activities [6,7,8]. Compound 3, built on the cGP scaffold, enhances serotonin receptor inhibition with an IC_50_ of 1.86 µM—two orders of magnitude better than the prototype. Other cGP-derived molecules, such as the D2R modulator (4) and CSP-2503 (5), demonstrate strong bioactivity, further establishing cGP as a key component in drug discovery.

The biosynthesis of cGP via biological processes is still a vital area of interest despite the development of synthetic methods for cGP scaffold [9,10]. Advances have been made using engineered *Escherichia coli* expressing specific enzymes, such as non-ribosomal peptide synthetases and cyclodipeptide synthases, to construct this specific peptide bond [11,12]. However, efficient biosynthesis from amino acids like proline and glycine continues to face major challenges related to precise peptide bond formation and selectivity [13]. This has spurred research into natural metabolic processes, particularly in marine fungi, which exhibit special physiological abilities in harsh conditions [14,15].

Mangrove-derived fungi, such as *Penicillium* and *Aspergillus* species, have emerged as promising sources of cGP due to their rich secondary metabolite production under the unique marine conditions [16]. These fungi offer a sustainable and possibly high-yield alternative for cyclic peptide production. However, to harness their full capacity, efficient screening of wild strains is necessary to facilitate gene cluster mining and identify biosynthetic pathways that can be optimized for industrial applications [17].

Advanced analytical techniques such as Selected Reaction Monitoring (SRM) have become essential to support this exploration [18]. The high-resolution mass spectrometry method provides the specificity, sensitivity, and precision needed to quantify small molecules in complex mixtures [19,20,21]. Using a triple-quadrupole mass spectrometer offers superior mass accuracy, resolution, and ease of data acquisition compared to traditional HPLC, making it an indispensable tool in metabolomics and natural product discovery.

In this study, we aim to develop a robust UPLC-MS/MS method utilizing an SRM method to quantify cGP in fungal fermentation extracts (Figure 2). This approach will be applied to *Penicillium* and *Aspergillus* strains to assess their biosynthetic capacity, offering new insights into sustainable cGP production and laying the foundation for upcoming biology and drug development applications.

## 2. Materials and Methods

### 2.1. Materials

Marine-derived fungal strains, including species of *Penicillium* and *Aspergillus*, were obtained from the low-temperature (−20 °C) strain storage of the Xiamen Key Laboratory of Marine Natural Product Resources, Xiamen Medical College, China. Cyclic glycine–proline (≥98% purity) was purchased from Bide Pharmatech Co. Ltd. (Shanghai, China). Chromatographic-grade chemicals such as methanol and ethyl acetate, were used throughout the study.

### 2.2. Fungal Strain Culture and Fermentation and Sample Preparation

A total of 15 fungal strains were utilized in this study. The strains were initially activated on sterilized Potato Dextrose Agar (PDA) and incubated at 28 °C for 4 days in sealed containers. Once the spores matured, approximately 1 cm³ of the culture was transferred to a 1 L Erlenmeyer flask containing 500 mL of sterilized Potato Dextrose Broth (PDB). The cultures were then incubated at 28 °C with shaking at 180 rpm for 7 days. Additionally, an uninoculated blank PDB medium was cultured as a parallel control group under the same conditions.

Following incubation, the fungal mycelium was separated from the culture broth and extracted three times with 200 mL of ethyl acetate. The combined organic phases were concentrated under reduced pressure, yielding crude ethyl acetate extract. Subsequently, ultrasound-assisted extraction of the mycelium was performed using 200 mL methanol for 30 min, followed by centrifugation and three additional extractions. The combined methanol extracts were evaporated to dryness, producing a crude methanol extract. The methanol residue was mixed with 100 mL of water and further extracted three times with 100 mL of ethyl acetate. The ethyl acetate layers and methanol residues were combined and concentrated to yield the crude metabolite. The blank control group underwent the same post-treatment procedure as described for the fungus extraction; the extract from the blank control group was also collected.

A 10 mg portion of crude extract from both the fermented samples and the blank control group was dissolved in chromatographic-grade methanol to prepare a test solution with a concentration of 20.0 μg/mL. Additionally, 10 mg of the cGP standard was dissolved in methanol to prepare standard solutions with a concentration range of 20–220 ng/mL. Both test and standard solutions were then filtered through a 0.22 µm microporous membrane into a small vial for subsequent UPLC-MS/MS analysis. Notably, the relative abundance values for the fungal extract were calculated by subtracting the blank control value from the detected peak area of the fungal extract.

### 2.3. UPLC-MS/MS Methods

In the SRM assay, a Shimadzu Nexera X2 LC-30AD UHPLC system, coupled with a Shimadzu triple quadrupole mass spectrometer (Kyoto, Japan), was used for the separation and detection of target metabolites. A Shim-pack XR-ODS Ⅲ column (75 × 2 mm i.d., 1.6 μm) was employed for the target dipeptide separation at 35 °C. Mobile phase A consisted of 0.1% formic acid in water, and mobile phase B in methanol. A linear gradient elution was used as follows: 0~4 min, 5~40% B; 4~4.5 min, 40~100% B; 4.5~6.5 min, 100% B; 6.5~6.6 min, 100~5% B; 6.6~10 min, 5% B, with a flow rate of 0.3 µL/min.

Key MS parameters included a spray voltage of 4Kv, gas flow rates (nebulizer: 2 L/min, heating: 10 L/min, dry gas: 10 L/min), interface temperature of 300 °C, and desolvation line (DL) temperature of 250 °C. For the product ion scan, the precursor ion was set at *m*/*z* 155.00, with a scan range of 20~200 and a time interval of 0.2 s.

### 2.4. Data and Statistical Analysis

All experiments were conducted in triplicate or sextuplicate, and results were expressed as mean ± standard deviation (SD). The relative standard deviation (RSD) was calculated to assess intra- and inter-day precision and spike-recovery. Calibration curves were generated across a concentration range of 20~220 ng/mL, with R^2^ > 0.999 indicating excellent linearity. Statistical significance was determined using GraphPad Prism 9.0, with a *p*-value of <0.05 considered significant.

## 3. Results and Discussions

### 3.1. Fungal Strain Growth and Morphology

The 15 marine mangrove-sourced fungal strains, comprising 7 *Aspergillus* and 8 *Penicillium* species, displayed distinct growth patterns on PDA (Figure 3). Colony diameters ranged from 20 to 45 mm, with textures varying from cottony to woolly and abundant aerial mycelia. Both genera formed mycelial pellets that were either spherical or oval, with minor morphological differences in shape and structure. However, *Aspergillus* strains were characterized by fast-growing dense mycelial networks, while *Penicillium* strains showed slower growth with more loosely branched mycelia [22].

The mature spores of each species were inoculated into sterilized media for incubation. After seven days of fermentation, crude extracts were obtained for subsequent cGP content analysis. Despite their physical similarities, these fungi from marine mangroves offer the potential to biosynthesize a diverse array of valuable secondary metabolites under controlled fermentation conditions.

### 3.2. Quantification and Monitoring of cGP

In SRM mode, the parent ion with the highest response was predefined. After undergoing collision-induced fragmentation in Q2, specific sub-ions were filtered in Q3 before being transmitted to the detector [23]. The peak areas of these transitions enabled precise analysis of target metabolites within complex secondary mixtures. Building on the research of Furtado [24], which characterized the secondary mass spectrum of cGP derivatives from *Aspergillus fumigatus*, it was confirmed that the main product ions of cGP differ significantly from those of its derivatives due to their unique fragmentation patterns. For example, the product ion at *m*/*z* 70 is almost absent in the derivatives, while a distinct ion at *m*/*z* 82 remains characteristic of cGP. This allowed us to leverage this highest response strength of ion fragments in SRM mode for effective quantification of cGP, avoiding interference from homologous compounds. For optimal conditions, we selected parent ion at *m*/*z* 155.05 with a Q1 pre-selection deviation of −0.012 Da and collision energies ranging from −21.0 to −22.0 kV. These conditions generated specific sub-ions at *m*/*z* 70.15 and *m*/*z* 82.10, which showed the highest response values and specificity for cGP quantification. The Q3 pre-selection deviations for these sub-ions ranged between −0.015 and −0.013 Da, with the ions designated as quantitative and qualitative ions, respectively. Additionally, alternative sub-ion fragments generated from the parent ion fragmentation, including *m*/*z* 58.05, *m*/*z* 99.15, and *m*/*z* 127.10, are proposed in their structures in Figure 4.

Qualitative monitoring of 15 wild-type marine mangrove fungal fermentation extracts using UPLC-MS/MS in SRM mode, based on the characteristic ions of cGP at *m*/*z* 70.15 and *m*/*z* 82.10, revealed the biosynthetic potential of these strains. While several strains demonstrated potential for cGP biosynthesis, *Penicillium incoloratum, Penicillium citrinum*, and *Aspergillus alabamensis* showed notable results. However, the highest abundance of cGP was observed in *Penicillium pedernalense* and *Penicillium steckii* (Figure 5).

### 3.3. Method Validation

The method was validated in line with FDA guidelines for bioanalytical method validation, focusing on sensitivity, linearity, precision, and accuracy. A highly specific, cost-effective SRM analysis method was developed using a triple quadrupole MS, targeting the quantitative characteristic ions of cGP at *m*/*z* 70.15.

The limit of detection (LOD) for cGP was determined to be 4.8 ng/mL, and the limit of quantification (LOQ) was 16.0 ng/mL, based on signal-to-noise ratios of 3 and 10, respectively (Figure 6a,b). Standard calibration curves were generated in SRM mode by analyzing standard dilutions across a linear concentration range of 20–220 ng/mL. Six concentration levels were analyzed in triplicate to quantify cGP based on relative abundance. The resulting regression equation for the standard curve was y = 3199x + 9472, with a correlation coefficient of R^2^ > 0.999 (*p* < 0.0001) (Figure 6c).

The method was also validated for both intra-day and inter-day precision. Intra-day precision (repeatability) was assessed by performing six repeated analyses (*n* = 6) of a standard solution at specified concentration levels within a single day, with precision expressed as the relative standard deviation (RSD) of the replicate measurements. (Table 1) Inter-day precision was evaluated over three separate days within one week to measure reproducibility. The RSD for the repeatability was 1.65% with a mean concentration of 67.45 ng/mL, while the RSD for inter-day precision was 1.54%, with a mean concentration of 67.61 ng/mL—both meeting the acceptance criteria (≤15% deviation). Additionally, recovery was assessed using spiked standard solutions at concentrations of 33.5 ng/mL and 67.0 ng/mL. The recovery ranged from 85.70% to 90.00%, with an RSD of 1.60% to 2.98%, indicating consistent performance across concentration levels. These validation results confirm that the method demonstrates excellent sensitivity, linearity, precision, and recovery, with no observed interference from endogenous components during ionization, further reinforcing the method’s reliability.

### 3.4. Results of cGP Content

The validated method demonstrated excellent performance, confirming its effectiveness for quantifying cGP content in marine mangrove-derived fungal extracts. This method was specifically applied to fermentation extracts from *Penicillium pedernalense* and *Penicillium steckii*, with the results summarized in Table 2. The cGP content in *Penicillium pedernalense* was found to be 67.45 ng/mL, while *Penicillium steckii* produced 31.71 ng/mL. In terms of production yield, *Penicillium pedernalense* exhibited a significantly higher cGP yield (29.31 mg/L) compared to *Penicillium steckii* (8.51 mg/L). Both *Penicillium pedernalense* and *Penicillium steckii* demonstrated productivity levels that are competitive with *filamentous* fungus strains, underscoring their feasibility for industrial applications.

Although wild-type fungal yields can be enhanced through genetic modification and fermentation optimization, the production levels of *Penicillium pedernalense* and *Penicillium steckii* demonstrate competitive and potentially superior productivity under the conditions tested in this study. Our findings suggest that *Penicillium pedernalense* and *Penicillium steckii* naturally produce cGP efficiently, even under unoptimized fermentation conditions. Given the challenges associated with the chemical synthesis of cyclic peptides, biosynthetic pathways utilizing marine fungi offer a promising alternative for sustainable production. Furthermore, the ability to produce significant quantities of cGP opens new avenues for its application in the pharmaceutical and nutraceutical industries.

## 4. Conclusions

The marine mangrove fungal strains exhibited distinct growth patterns, with *Aspergillus* forming fast-growing, dense mycelial networks, while *Penicillium* displayed slower growth with loosely branched mycelia. Following fermentation, crude extracts were obtained for cGP content analysis, which illustrated the capability of marine-derived fungi from mangroves for the production of valuable secondary metabolites.

This work effectively established and validated a UPLC-MS/MS method for the quantification of cyclic glycine–proline (cGP) in fungal extracts obtained from mangroves. The method demonstrated excellent sensitivity, precision, and recovery, confirming its reliability for studying cyclic peptide biosynthesis. Both tested fungi produced significant amounts of cGP, with P. *pedernalense* having a superior yield of 29.31 mg/L, demonstrating their feasibility for sustainable production and promising industrial applications in drug development and biotechnology [25].

Future research could focus on optimizing fermentation parameters and employing genetic engineering techniques to enhance cGP yields in these strains [26]. Also, studying other fungi that grow in mangroves might help find new sources of cGP or related bioactive compounds, which would help create more environmentally friendly ways for pharmaceuticals. The significance of this study lies in its contribution to sustainable marine biotechnology, particularly through quantitative metabolite tracking for precise strain screening, which can be employed for high-value compound production while eliminating the petrochemical consumptions for conventional synthetic approaches.

## Figures and Tables

**Figure 1 jof-10-00779-f001:**
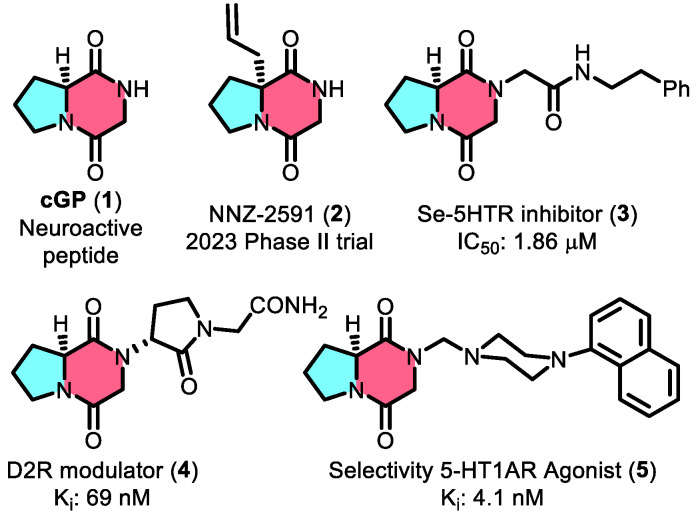
The scaffold of cyclic glycine–proline and its representative applications in drug discovery.

**Figure 2 jof-10-00779-f002:**
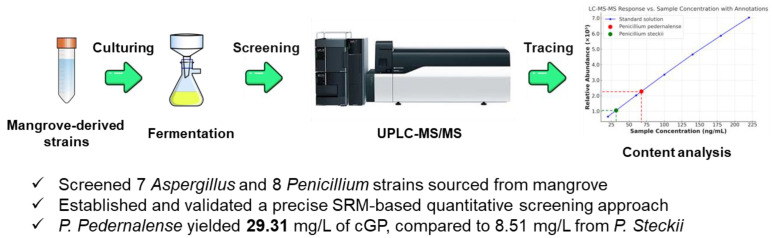
Simplified workflow for efficient screening of mangrove-derived strains using a quantitative UPLC-MS/MS-SRM approach to measure cyclic glycine–proline content.

**Figure 3 jof-10-00779-f003:**
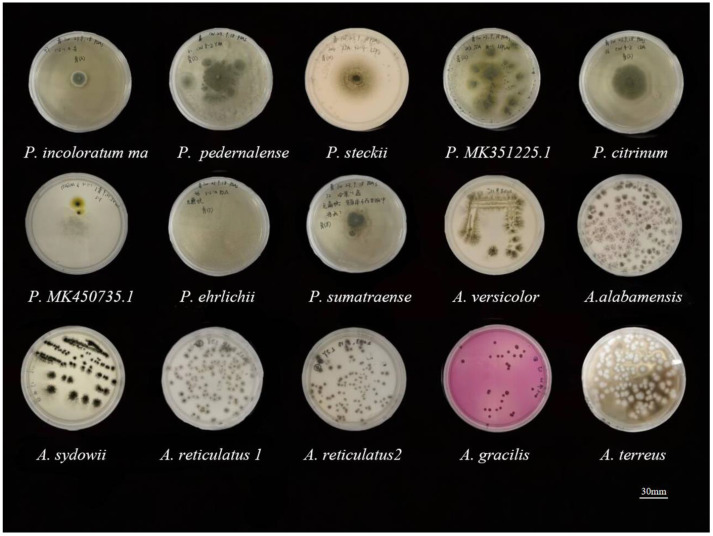
Morphological comparison of *Penicillium* sp. and *Aspergillus* sp.

**Figure 4 jof-10-00779-f004:**
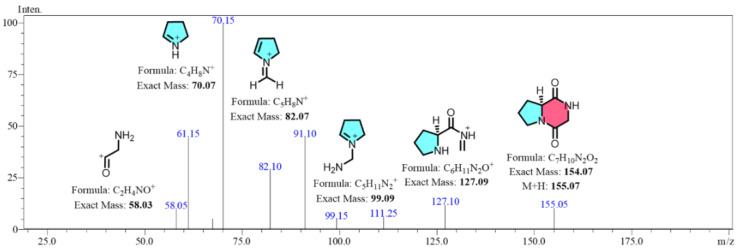
Full scan mass spectrometric data of the cGP standard, highlighting the parent ion at *m*/*z* 155.05 and its characteristic product ions at *m*/*z* 70.15 and 82.10, along with alternative sub-ion fragments generated from parent ion fragmentation.

**Figure 5 jof-10-00779-f005:**
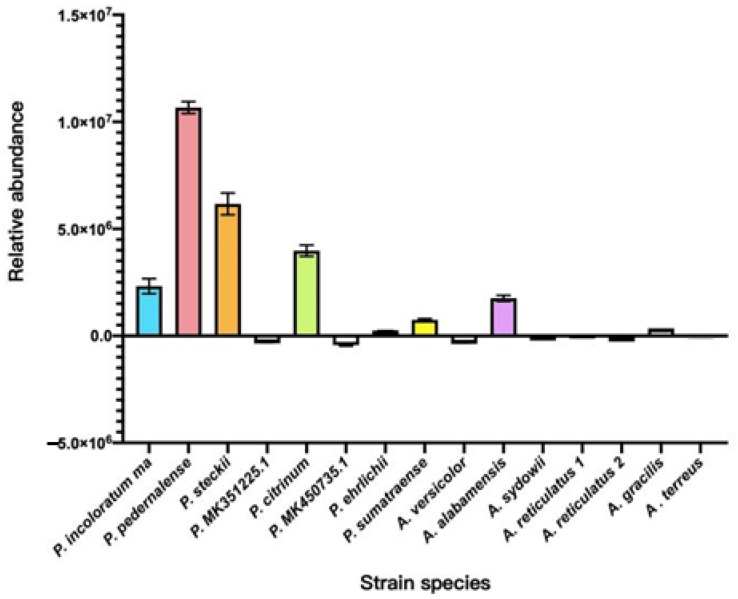
High-efficiency screening of wild-type fungal strains from marine mangroves using UPLC-MS/MS-SRM (Note: Relative abundance is calculated as the detected peak area minus the blank control. A negative value indicates a lower peak area than the control, suggesting a weaker biosynthetic activity).

**Figure 6 jof-10-00779-f006:**
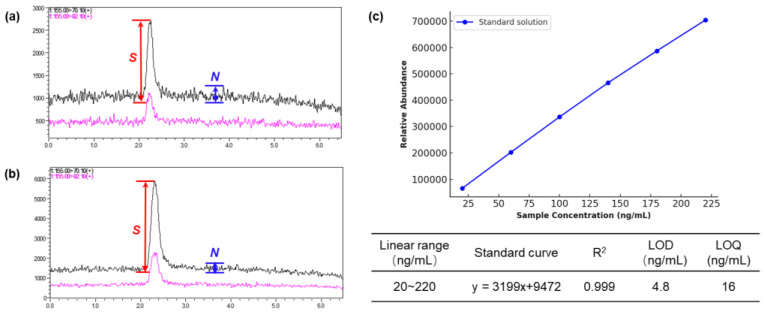
(**a**) LOD and (**b**) LOQ determination for cGP quantification in SRM mode, and (**c**) Calibration curve of standard solution. Note: The black signal line in the left figures represents the monitoring result of the quantitative ion of cGP at *m*/*z* 70.10, while the pink signal line corresponds to the monitoring of the qualitative ion of cGP at *m*/*z* 82.10. The signal-to-noise ratio (*S*/*N*) is calculated as the signal (*S*) divided by the noise (*N*). Detailed values are provided in Appendix A in the Appendix A.

**Table 1 jof-10-00779-t001:** Validation of method by intra-day and inter-day precision and recovery. Precision was evaluated with a cGP standard solution at 20 μg/mL.

Repeatability (% RSD, *n* = 6)		Precision (% RSD, *n* = 6)		Recovery (%, *n* = 6)			
				33.5 ng/mL		67.0 ng/mL	
Mean (ng/mL)	RSD	Mean (ng/mL)	RSD	Mean	RSD	Mean	RSD
67.45	1.65	67.61	1.54	88.62	2.90	90.04	1.60

**Table 2 jof-10-00779-t002:** Results of cGP content and corresponding fermentation yield.

Sample	Measured Content (ng/mL, Mean ± SD)	Production (mg/L, Mean ± SD)
*Penicillium pedernalense*	67.45 ± 1.11	29.31 ± 0.61 ^a^
*Penicillium steckii*	31.71 ± 0.31	8.51 ± 0.15 ^a^
Blank control	14.15± 0.16	--

Note: ‘a’ refers to the fermentation yield calculated based on the actual content, determined by subtracting the blank control content from the measured content.

## Data Availability

The original contributions presented in the study are included in the article and Appendix A, further inquiries can be directed to the corresponding authors.

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
