# Peer review of "Quantitative Monitoring of Cyclic Glycine–Proline in Marine Mangrove-Derived Fungal Metabolites"

_jof, 2024, doi:10.3390/jof10110779_

Round 1
Reviewer 1 Report
Comments and Suggestions for Authors
Please see the attachment.

Author Response
- There are several points to be addressed by the authors. 1. From reading the Abstract as the self-explanatory section, the sentence "When applied to fungal extracts, Penicillium pedernalense achieved 67.45±1.11 ng/mL (yielding 29.31±0.61 mg/L), surpassing Penicillium steckiiat 31.71±0.31 ng/mL (yielding 8.51±0.15 mg/L)." is not clear. What are the values in parentheses?
We sincerely thank the reviewer for this valuable suggestion. We have rephrased the sentence substantially to enhance the clarity of the abstract.
- Two paragraphs in the subsection 3.2. "Monitoring and quantification of cGP" are not organized sequentially. The first paragraph is dealt with the product ions at m/z 70 and m/z 82. The origin of just these values seems unclear to (several or even many) readers. By interchanging the order of the first and second paragraphs, it will become clear where these quantities originate, and what conditions generated specific sub-ions at m/z 70.15 and m/z 82.10. Whereas the content of the Lines 156-159, and the Figure 4, could remain at their prior position.
Paragraph reorganized: See the lines 147-161.
- "Materials and Methods": "blank control was extracted using the same post-treatment" (Line 99). Please point out what subject served as a blank control. It is all the more important that the relative abundance values are calculated as the detected peak area minus the blank control.
Sentence rephrased: See the lines 91-92, 101-103,109-111
- Materials and Methods": "reference extract was dissolved …" (Line 101). Was it a commercial cyclic glycine-proline? I suppose it should not be named "extract".
The term "extract" derived from fermented samples and the blank control group, rather than a commercial cyclic glycine-proline. We have rephrased this section for clarity—please refer to lines 104-105 for the updated wording.
- Figure 2 is not referred to in the text. Figure 3 (A) is not referred to in the text, and Figure 3 (B) is referred to as Figure 3, "Flow chart of extraction …".
Corrected in lines 70 and 134, respectively.
- Two "Table 1", one is "Results of cGP content and corresponding fermentation yield", and second is "Method validation by intra-day and inter-day precision, and recovery. Precision was assessed using a standard solution with concentrations spanning the entire calibration range", both being not referred to in the text, supplemented by mentioning "Table 4" at Line 215 (there are only 2 tables in the manuscript), require formal correction.
Corrected the lines 200-201 and 225.
- Selected Reaction Monitoring (SRM), once abbreviated at Line 63, should be used in the abbreviated form in the rest text (see Lines 106, 146).
Corrected
- The term "fungals" in the context "Both tested fungals produced significant amounts …" (Line 247) should be replaced by "fungi". Other corrections to manuscript 10. Line 53: "from amino acids like protein and glutamine" should be replaced by "from amino acids like proline and glutamine".
The paragraph "from amino acids like protein and glutamine" has been rephrased to enhance clarity and readability.

Reviewer 2 Report
Comments and Suggestions for Authors
Lin et al. report on the development and validation of a robust UPLC-MS/MS method for quantification of cyclic glycine-proline (cGP) produced by mangrove0derived Aspergillus and Penicillium strains. cGP, a widely studied marine-derived cyclic peptide, is a building block for a number of preclinical candidates. Development of a purification scheme form biological sources would be advantageous versus synthetic methods. In this manuscript, high resolution mass spectrometry technique developed to quantify cGP in fungal fermentation extracts showed great promise in precision, recovery, and its low detection limits.
Specific Comments:
1. Table 1 is not cited in the manuscript
2. A Table 4 is cited in the manuscript. This citation should be to the second Table 1 in the manuscript, which should be renumbered as Table 2.
Comments on the Quality of English Languagenone specific
Author Response
Thank you for your valuable comments. We appreciate your feedback regarding the citation of Table 1 and the reference to Table 4. We will ensure that Table 1 is appropriately cited in the manuscript and correct the numbering by renumbering it as Table 2.

Reviewer 3 Report
Comments and Suggestions for Authors
Major points
1. Although authors described the importance of DKP for drug leads (line 27-43) in Introduction, they focused on only cGP to identify and quantify it in fungal cultures by UPLC-MS/MS. Recent works are focusing on establishment of molecular network of structurally related natural compounds using UPLC-MS/MS. In this point, this manuscript is too limited, so readers are not so much interested in this paper.
2. Authors isolated mangrove-derived fungi to quantify the cGP in the cultures. If they emphasize the importance of marine-derived fungi as a drug discovery resource as described in Introduction, they should compare the cGP amounts in the cultures of terrestrial fungi.
3. Authors described that marine/mangrove-derived fungi produced rich secondary metabolite under marine condition (line 57-59), but they culture fungi under normal conditions (line 85-90). Any reason?
Minor points
4. line 137-139; what are "These findings"? Afraid that readers cannot understand this logic.
5. line 106; correct to "Selected Reaction Monitoring (SRM)". After this, SRM should be used (line 146).
6. line 215; correct Table 4 to Table 2.
7. line 222; correct Table 1 to Table 2.
8. cGP solution is in methanol?
9. Delete Figure 1b), that is not necessary.
10. What is the brand of MS apparatus ? Does Shimadzu system include MS part ?
11. line 180-186; Readers cannot understand how LOD and LOQ are calculated from Figure 6 in which the words (black and red) are too small.
Author Response
- Although authors described the importance of DKP for drug leads (line 27-43) in Introduction, they focused on only cGP to identify and quantify it in fungal cultures by UPLC-MS/MS. Recent works are focusing on establishment of molecular network of structurally related natural compounds using UPLC-MS/MS. In this point, this manuscript is too limited, so readers are not so much interested in this paper.
Thank you for this insightful feedback. We acknowledge that this study may indeed appeal to a specialized audience due to its targeted focus on quantifying cyclic glycine-proline (cGP) in mangrove-derived fungi. While broader molecular networking approaches, like GNPS, are invaluable for identifying diverse novelty natural compounds, our study offers a precise detection method to track cGP, an industrially valuable DKP molecular, from possible sustainable microbial sources for applications in healthcare and pharmaceuticals.
We understand that cGP, as a relatively simple molecule, may seem less novel to experts in natural product chemistry who often seek novel chemical entities for drug discovery. However, as an endogenous molecule with unique ADMET properties and bioactivity through protein-protein interactions, cGP holds significant potential for therapeutic research in CNS related disorders. Our method aims to facilitate future studies on its sustainable biosynthesis and possible applications in drug development.
- Authors isolated mangrove-derived fungi to quantify the cGP in the cultur If they emphasize the importance of marine-derived fungi as a drug discovery resource as described in Introduction, they should compare the cGP amounts in the cultures of terrestrial fungi.
Thank you for this constructive comment on comparising of cGP content in marine-derived versus terrestrial fungi. This is indeed a valuable scientific question that warrants further investigation. Currently, there is limited data on cGP levels in terrestrial fungi, as most research has focused on bioactivity screening rather than quantification. While our department primarily focuses on marine-derived fungi, collaborating with terrestrial fungi research group could allow us to explore this meaningful comparison. Such a study, supported by genomic analysis, could provide insights into how environmental factors influence microbial metabolite production. We appreciate your suggestion and consider this a promising direction for future research.
- Authors described that marine/mangrove-derived fungi produced rich secondary metabolite under marine condition (line 57-59), but they culture fungi under normal conditions (line 85-90). Any reason?
Fungi from marine and mangrove environments are known to produce a diverse range of secondary metabolites under natural marine conditions. For our initial screening, we cultured these fungi in standard laboratory conditions, without fully replicating natural seawater parameters like salinity, to assess their baseline metabolite production. In future experiments, we plan to optimize fermentation conditions by simulating marine-like environments more closely to determine whether these adjustments can enhance cGP production.
- line 137-139; what are "These findings"? Afraid that readers cannot understand this logic.
The paragraph has been rephrased to enhance clarity and readability. See the lines 142-146.
- line 106; correct to "Selected Reaction Monitoring (SRM)". After this, SRM should be used (line 146).
Corrected.
- line 215; correct Table 4 to Table 2.; line 222; correct Table 1 to Table 2.
Corrected.
- cGP solution is in methanol? Delete Figure 1b), that is not necessary.
The cGP standard and fungal fermentation extract were dissolved in chromatographic-grade methanol, and Figure 1b has been deleted as requested.
- What is the brand of MS apparatus ? Does Shimadzu system include MS part ?
Sentence rephrased. See the lines 113-114.
- line 180-186; Readers cannot understand how LOD and LOQ are calculated from Figure 6 in which the words (black and red) are too small.
Figures 6a and 6b have been updated, and additional notes have been added to the figure captions to clarify the calculation of LOD and LOQ.

Round 2
Reviewer 3 Report
Comments and Suggestions for Authors
Thank you for responding to my comments. I understood your revision. Although I think you can make this research better, this manuscript is acceptable for this journal.